# Single or Daily Application of Topical Curcumin Prevents Ultraviolet B-Induced Apoptosis in Mice

**DOI:** 10.3390/molecules28010371

**Published:** 2023-01-02

**Authors:** Khairuddin Djawad, Ilham Jaya Patellongi, Upik Anderiani Miskad, Muhammad Nasrum Massi, Irawan Yusuf, Muhammad Faruk

**Affiliations:** 1Department of Dermatology and Venereology, Faculty of Medicine, Hasanuddin University, Makassar 90245, Indonesia; 2Department of Physiology, Faculty of Medicine, Hasanuddin University, Makassar 90245, Indonesia; 3Department of Pathological Anatomy, Faculty of Medicine, Hasanuddin University, Makassar 90245, Indonesia; 4Department of Microbiology, Faculty of Medicine, Hasanuddin University, Makassar 90245, Indonesia; 5Department of Surgery, Faculty of Medicine, Hasanuddin University, Makassar 90245, Indonesia

**Keywords:** curcumin, radiation, apoptosis

## Abstract

Curcumin is a natural ingredient with antioxidant effects, widely studied as a treatment for various types of cancer. However, its effects on ultraviolet radiation have not been fully explored. The effects of single or daily application of 0.1–100 μM curcumin on cell apoptosis in ultraviolet B (UVB)-induced mice were tested using an experimental double-blind posttest design with a control group and two research models: a single application of curcumin before a single UVB exposure and daily application of curcumin for 7 days before a single UVB exposure on the seventh day. Apoptotic cells were counted using a tunnel system kit. The number of apoptotic cells under a single or daily application of curcumin for 7 days was significantly lower than that of the UVB controls (*p* ≤ 0.05). The number of apoptotic cells decreased with the increasing concentration of curcumin, and the maximum effect was observed at 100 μM. Daily application of topical curcumin was superior in preventing apoptosis (mean apoptotic cell count of 14.86 ± 1.68) compared with a single application (17.46 ± 0.60; *p* = 0.011). Topical curcumin can act as a potential photoprotective agent in preventing cutaneous malignancies due to UVB radiation. Further studies are warranted, especially in humans.

## 1. Introduction

Curcumin is a polyphenolic compound extracted from turmeric (*Curcuma longa)* and is a yellowish pigment that contains various metabolites, such as dihydrocurcumin, tetrahydrocurcumin, hexahydrocurcumin, octahydrocurcumin, curcumin glucuronide, and curcumin sulfate. This compound has been used as a chemotherapeutic or chemopreventive agent and an alternative therapy for various diseases [1]. Curcumin possesses antioxidant, anti-inflammatory, immunomodulatory, hepatoprotective, anti-ischemic, nephroprotective, antimicrobial, hypoglycemic, and antirheumatic properties [2]. It also has anticarcinogenic effects related to its molecular activity of inducing the apoptosis of cancerous cells but interestingly sparing healthy cells. This compound has been used in the therapy of breast, pancreatic, and colorectal cancers, and multiple myeloma [1,3]. 

Topical curcumin at a dose of 30 µM can induce apoptosis in several tumor cells, thereby reducing cell cycle progression and preventing the growth of cancer cells [4]. The apoptotic effect of curcumin depends on the concentration and duration of its administration. In terms of morphology, the apoptotic cells exhibit several characteristic changes compared with normal cells, such as shrinkage, pyknosis, and plasma membrane blebbing, leading to the formation of apoptotic bodies [1,5]. In contrast to its effect on cancerous cells, curcumin inhibits the apoptosis of keratinocyte cells in ultraviolet-induced cells [6]. However, the research on its influence on ultraviolet B (UVB) radiation is still limited. Previous studies used keratinocytes as the main target of UVB exposure to explore photodamage and apoptosis [7] and found that curcumin can prevent the accumulation of the abnormal cells that cause skin cancer [7,8,9]. The application of curcumin prior to UVB exposure in mice suppressed erythema and apoptosis by inhibiting p53 and interleukin-6 (IL-6) expression and increasing anti-inflammatory IL-10 expression in the pretreated skin compared with those in the controls [8]. Information regarding the anti-apoptotic effect of topical curcumin is limited, especially in UVB-induced cell apoptosis.

This study aimed to explore the preventive effect of topical curcumin on UVB-induced mice by histopathologically examining the number of apoptotic cells using two different study designs. The first group was applied with topical curcumin in four concentrations once, followed by a single UVB exposure. The second group was applied with topical curcumin in four concentrations once daily for 7 consecutive days, followed by a single UVB exposure. Biopsy was then performed to determine the number of apoptotic cells. The optimal number of applications and concentration were also determined for the potential use of topical curcumin as a photoprotective agent.

## 2. Results

The weight of all mice was homogenous and did not differ significantly across all test groups (*p* = 0.76). Thus, the difference in the effect of all topical curcumin concentrations for all the treatment groups could be further analyzed. ANOVA showed an apoptosis-inhibitory effect in both research models (*p* ≤ 0.05). Post hoc analysis revealed that the number of apoptotic cells in the non-UVB-exposed group was significantly different from that in the UVB and acetone controls (*p* ≤ 0.05). All groups that received the four concentrations of topical curcumin in either single or daily applications with subsequent UVB exposure showed significantly lower apoptotic cell counts compared with the UVB control group (*p* ≤ 0.05). Significant differences in apoptotic cell count were observed among the curcumin groups with various concentrations (*p* ≤ 0.05), except for the 1 μM group (*p* > 0.478). A similar inhibitory effect against UVB-induced apoptosis was noted in the second experimental model (Figure 1).

Table 1 exhibits the difference in apoptotic cell count between the single application of topical curcumin of various concentrations with a subsequent single UVB exposure and the daily application of topical curcumin for 7 days with a subsequent single UVB exposure. The *t*-test showed a statistically significant difference in apoptotic cell count between the one-time and seven-time curcumin applications among all concentration groups (*p* < 0.05) except the 1 μM group. All curcumin concentrations in a single application resulted in a decrease in the apoptotic cell count. The number of cells was similar among the various concentrations except for 100 μM. When curcumin was applied daily for 7 consecutive days, the number of apoptotic cells decreased significantly under treatment concentrations of 10 μM and 100 μM with a mean of 21.73 ± 1.64 and 14.86 ± 1.68, respectively (*p* = 0.041, 0.011). At low concentrations of 0.1 μM and 1 μM, the single application of curcumin (mean number of apoptotic cells: 27.86 ± 1.78 and 27.76 ± 0.96, respectively) was superior to the daily application (30.70 ± 2.23 and 29.33 ± 5.57, respectively). However, only the results for 0.1 μM were statistically significant (*p* = 0.042).

Histopathological examination showed that regardless of the number of applications, an increase in curcumin concentration led to a decrease in cell apoptosis compared with the positive control. This finding revealed the inhibitory effect of topical curcumin, which was most prominent at the 100 μM concentration after the seven-time application (Figure 2A,B).

Based on histopathological examination, with an increase in curcumin concentration, both in single and daily application, the results showed a decrease in cell apoptosis compared to the positive control. This showed the inhibitory effect of topical curcumin, most prominently in the 100 μM concentration after the seven-time application (Figure 2A,B).

## 3. Discussion

Cell apoptosis or programmed cell death serves as an integral part of cellular homeostasis. An increase in the abnormal viability of cells influenced by endogenous or exogenous factors can lead to the development of various diseases, most prominently malignancies and autoimmune diseases. The protein family BCL-2 is responsible for regulating cellular apoptosis balance through pro-apoptotic and pro-survival members and thus serves as a basis for potential therapeutic developments for various diseases [10]. Cell apoptosis occurs through two pathways; extrinsic or cytoplasmic pathways triggered by Fas death receptors and intrinsic or mitochondrial pathways triggered by the release of cytochrome C from mitochondria. One of the most important proteins in this pathway is BCL-2. The overexpression of this protein leads to cell accumulation in the G0 phase [11]. Although apoptosis is a defense mechanism to prevent cell mutations that can lead to malignancies, the prevention of cell damage or repairing of cell damage through normal cell mechanism in cases of mild radiation is preferred [12,13].

The usefulness of curcumin has long been explored, and the compound is associated with modulation on various pathways. Special interest has been directed to the autophagy or degradation of supernumerary or dysfunctional components within cells, in which curcumin inhibits the formation of reactive oxygen species (ROS) and acts as an antioxidant agent [14,15]. The topical use of curcumin is ideal compared with systemic administration; even though the compound has a high safety profile, it is poorly absorbed systemically and has a rapid elimination [16]. Curcumin can either induce or inhibit cellular apoptosis in various malignant cells and concentrations, such as in human melanoma cells (30–60 mM for 24 h), human leukemia (HL) 60 cells (10–40 mM for 16–24 h), AK-5 tumor cells (10 mM for 18 h), and MCF-7 breast cancer cells (25 mM for 24 h). Apoptosis inhibition was observed in dexamethane-induced apoptosis in rat thymocytes and chemotherapy-induced apoptosis in breast cancer cells [17]. An evaluation of the efficacy of curcumin for treating infantile hemangioma endothelial cells (HemECs) found that its 100 μM concentration exhibited a high inhibition activity for the proliferation capability of HemECs; as proven by positive annexin-V-FITC staining, caspase-3 activation, and the cleavage of poly(adenosine diphosphate-ribose) polymerase (PARP) in the treated cells, curcumin achieved low-micromolar IC50 (half maximal inhibitory concentration) and induced apoptosis in HemECs through the downregulation of myeloid cell leukemia-1 (MCL-1) and hypoxia-inducible factor 1a [18]. Furthermore, curcumin is useful in treating psoriasis by reducing epidermal thickness, erythema, pruritus, and burning and pain sensations [19]. 

Minimal differences in apoptotic cell count were observed among the three curcumin concentrations of 0.1, 1, and 10 μM in single application with subsequent UVB radiation. However, a drastic decrease was found when the concentration was increased to 100 μM. In addition, the highest concentration (100 μM) yielded the lowest number of apoptotic cells after daily use for 7 consecutive days, providing the best result in this study. Although an increase in application and concentration seemed to result in an overall low apoptotic cell count, the single application of the low concentrations of 0.1 and 1 μM was superior compared with their daily application. We have yet to establish the reason behind this phenomenon. However, we hypothesized that this phenomenon may be related to the biphasic effect of curcumin on the oxidation of post-prandial chylomicrons and its biphasic hormetic response on proteasome activity and heat-shock protein synthesis in human keratinocytes [20,21]. Furthermore, we postulated that the cumulative effect of antioxidants during the daily application of topical curcumin enhances the inhibitory effect on cell damage and apoptosis, as reported by Zhou et al. [22]. Another study regarding the pre-treatment of oral curcumin during a seven-day period prevented cyclophosphamide-induced lung injury in rats through the suppression of oxidative stress, therefore reducing the number of cell apoptosis [23].

The protective effect of curcumin in this research was supported by other experimental studies. Li et al. [16] reported that pretreatment with curcumin effectively inhibited photodamage in mice and human keratinocyte (HaCaT) cells using a UVB dose of 540 mg/cm^2^ (3 MED) for 3 consecutive days. Different results were obtained by Park et al. [24], who also pretreated HaCaT cells with curcumin before exposing them to a subapoptotic dose of UVB (100 mJ/cm^2^) to induce apoptosis. The difference in results may be due to the differences in UVB doses. The pretreatment of curcumin prior to UVB radiation can reduce the number of ROS by acting as a scavenger for most ROS [14,16]. Although the production of ROS after UVB radiation is a natural and protective response toward UVB radiation, the number of ROS released should be controlled or reduced because an excessive amount of oxidative stress can damage various cellular compounds, such as nucleic acids, proteins, and lipids. It can lead to a gene mutation that acts as a precursor for photoaging and cutaneous malignancies [25]. Reactive oxygen species also play an important role in UVA damage. A study evaluating the efficacy of pretreatment with topical curcumin before UVA radiation found that this compound significantly decreased the level of NF-κB, a protein that plays an integral role in the inflammation pathway [26]. Furthermore, curcumin has moderate inhibition properties toward proteins MMP-1 and MMP-3, which are expressed after UVA and UVB radiation, resulting in the restoration of collagen metabolism, wound healing, and regulating disorders [26,27]. Recently, Li et al. [16] and Barbalho et al. [19] found that curcumin prevents apoptosis by inhibiting the production of ROS through the modulation of the Nrf2 and NF-κB signaling pathways. Nrf2 is a key transcription factor in oxidative stress that can influence the activation of antioxidant response by various cytoprotective and antioxidant enzyme genes [6,28,29,30].

In the current study, we observed that the application of topical curcumin gave a protective effect against ultraviolet exposure-induced apoptosis in mice. However, this study has limitations because it did not examine the markers of oxidative stress (reactive oxygen species, hydrogen peroxide, and malondialdehyde) and inflammation (cyclooxygenase-2, interleukin, prostaglandin E2, tumor necrosis factor-alpha, and nitric oxide) that may have an effect on the process of apoptosis.

## 4. Methods

### 4.1. Study Design and Subject

The study was a double-blind experimental posttest design with a control group, conducted at the Hasanuddin University Animal Laboratory, Makassar, Indonesia. Healthy male Swiss albino mice aged 6–9 weeks and weighing 20–30 g were maintained under a temperature of 28 °C ± 2 °C and humidity of 50% ± 10% for a minimum of 1 week. All mice were shaved on the back regularly during the research.

The topical preparation was curcumin purchased from Sigma-Aldrich, Inc. (St Louis, MO, USA). It was diluted in acetone at concentrations of 0.1 μM, 1 μM, 10 μM, and 100 μM. The UVB lamp source was 10 FS-40-T12 fluorescent sun lamps with a spectrum of 280–340 nm and a peak emission of 314 nm. The UVB lights were calibrated with a FLUX radiometer.

### 4.2. Study Protocol

Two experimental models were used: a single application of various curcumin concentrations before a single UVB exposure and daily application of various curcumin doses for 7 days before a single UVB exposure on the seventh day. The mice were randomly allocated to 11 groups, each containing five mice. Group 1 received no treatment, group 2 received only UVB irradiation, and group 3 received acetone and UVB irradiation. Groups 4–7 received topical curcumin applications with concentrations of 0.1 µM, 1 µM, 10 µM, and 100 µM, respectively, with a dose of 2 µL/cm^2^ on the back for 20 min before exposure to 343 mJ of UVB. Groups 8–11 received topical curcumin applications at similar concentrations once daily for 7 days and were irradiated with 343 mJ of UVB 20 min after the last topical curcumin application on the seventh day. Skin biopsy for histopathological examination was performed 24 h after UVB exposure, and the obtained samples were assessed under 400× magnification.

### 4.3. Cell Apoptosis Activity Examination

The apoptotic cells among the squamous epithelial cells in all epidermis layers were counted histopathologically using the Apo-BrdU-IHC Kit TUNEL System (Biovision^®^, Milpitas, CA, USA), following the manufacturer protocol. After being deparaffinized and rehydrated, the tissue samples were permeabilized with proteinase K (30 μg/mL) for 20 min at room temperature and then removed from endogenous peroxidase activity using 3% hydrogen peroxide for 5 min. After being rinsed with phosphate-buffered saline (PBS), the samples were incubated with 5× reaction buffer for 10 min, added to DNA labeling solution, and incubated at 37 °C for 1.5 h. The reaction was stopped by immersing the tissue in PBS twice within 15 min. An anti-BrdU-Biotin solution was added to each slide and incubated for 1.5 h. The samples were washed with PBS and incubated with streptavidin-HRP for 30 min at room temperature before being rinsed again. After rinsing, the tissues were incubated with diaminobenzidine for approximately 10 min until a change to a brownish color was observed. The tissues were then rehydrated, immersed in xylol, covered with object glass, and examined under a microscope (Olympus Type CX-31, Tokyo, Japan). The number of apoptotic cells was counted by two independent observers.

### 4.4. Data Analysis

Data analysis was conducted using SPSS 18.0 for Windows (SPSS Inc. Chicago, IL, USA). The statistical tests used were one-way ANOVA for comparison between groups in each experimental design and the *t*-test for comparison between experimental designs. *p* ≤ 0.05 was considered statistically significant.

## 5. Conclusions

Topical curcumin can act as a photoprotective agent by preventing cell apoptosis in UVB-induced mice. In terms of daily application and increasing concentration, 100 μM curcumin treatment was associated with the smallest number of histopathologically observed apoptotic cells after a single UVB exposure. However, at low concentrations, a single application of curcumin was more beneficial than daily application. Further studies, especially using concentrations lower than 0.1 μM, are warranted to prove this hypothesis.

## Figures and Tables

**Figure 1 molecules-28-00371-f001:**
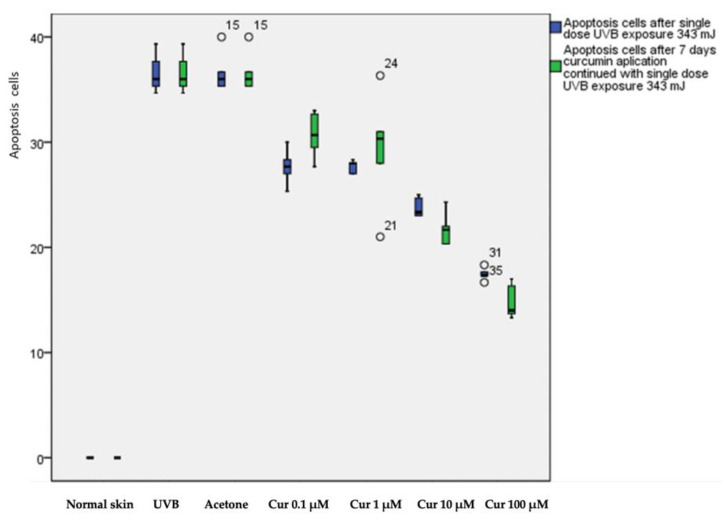
Induction of apoptosis in UVB-irradiated mice. Pretreatment with topical curcumin before UVB irradiation prevented apoptosis in both experimental models.

**Figure 2 molecules-28-00371-f002:**
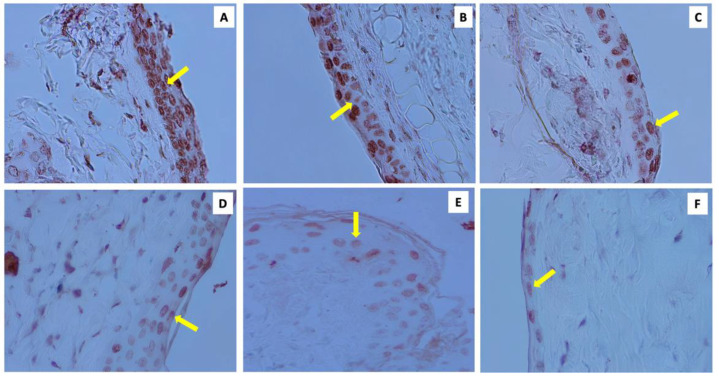
Apoptotic cells, indicated by brown staining in the nuclei (arrow), at 400× magnification under an Olympus Type CX-31 microscope, in the UVB-only group with majority apoptosis cells (**A**), acetone (**B**), 0.1 µM concentration (**C**), 1µM concentration (**D**), 10 µM concentration (**E**), and 100 µM concentration with the least amount of cell apoptosis (**F**).

**Table 1 molecules-28-00371-t001:** Comparison of apoptotic cell count between one-time and seven-time curcumin application before single 343-mJ UVB exposure.

Curcumin		Number of Apoptotic Cells		*p*-Value
Concentration	Number of Applications	Mean ± SD	Mean Difference	
0.1 μM	1	27.86 ± 1.78	−2.84	0.042 *
	7	30.70 ± 2.23		
1 μM	1	27.76 ± 0.96	−1.57	0.478
	7	29.33 ± 5.57		
10 µM	1	23.80 ± 0.96	2.07	0.041 *
	7	21.73 ± 1.64		
100 μM	1	17.46 ± 0.60	2.60	0.011 *
	7	14.86 ± 1.68		

Note: * Significant; SD, Standard deviation.

## Data Availability

The datasets generated during and/or analyzed during the current study are available from the corresponding author on reasonable request.

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
