# Peer review of "Single or Daily Application of Topical Curcumin Prevents Ultraviolet B-Induced Apoptosis in Mice"

_molecules, 2023, doi:10.3390/molecules28010371_

Round 1

Reviewer 1 Report (New Reviewer)

Curcumin is the biphenolic active compound  and has been used for hundreds of years to treat various disease. The role of curcumin and its antioxidant activity is well documented.  So this study seem quite interesting. Well the manuscript is written finely. So i recommend a minor revision as 
Introduction : page -2, "Information about the optimal concentration of 56 curcumin with .................................. of increasing the preventive effect. " This paragraph can be rewritten .
Methods:  "a true experimental"   What the meaning of true.

"Topical synthetic curcumin lotion 75 with concentrations of 0.1, 1, 10, and 100 μM were obtained from the Phytochemical Phar- 76 macognosy Laboratory, Faculty of Pharmacy, Hasanuddin University, M"

As the preparation  of curcumin used was obtained from some laboratory from its own university, content of lotion should be explain.
Detail of procurement of curcumin like company , city etc .
as a limitation of the study , that this study can be planed slightly better or studies can be supported with better reference .
In histopathological Fig 2A-B, it will be better if author can add effect after every treatment.
Arrow in the fig should be labeled properly 
Rest overall manuscript seems ok to me . 
thanks good luck 

Author Response

REVIEWER 1

Curcumin is the biphenolic active compound  and has been used for hundreds of years to treat various disease. The role of curcumin and its antioxidant activity is well documented.  So this study seem quite interesting. Well the manuscript is written finely.

So I recommend a minor revision as:

  1. Introduction : page-2, "Information about the optimal concentration of 56 curcumin with .................................. of increasing the preventive effect. " This paragraph can be rewritten.

Response:

We have revised introduction section on page 2, line 58-59.

Information regarding the anti-apoptotic effect of topical curcumin is limited, especially in UVB-induced cell apoptosis.

  1. Methods:  "a true experimental"   What the meaning of true.

Response:

We have revised the methods section on page 2, line 71.

“The design of the study was a double-blind experimental posttest design…”

  1. "Topical synthetic curcumin lotion 75 with concentrations of 0.1, 1, 10, and 100 μM were obtained from the Phytochemical Pharmacognosy Laboratory, Faculty of Pharmacy, Hasanuddin University, Makassar, Indonesia"

As the preparation  of curcumin used was obtained from some laboratory from its own university, content of lotion should be explain. Detail of procurement of curcumin like company, city etc.

Response:

We have revised the methods section on page 2, line 76-77.

“The topical preparation was curcumin that was purchased from Sigma-Aldrich, Inc. (St Louis, MO, USA). It was diluted in acetone with concentrations of 0.1, 1, 10, and 100μM.”

  1. As a limitation of the study, that this study can be planed slightly better or studies can be supported with better reference.

Response:

We have revised the discussion section on page 7, line 247-252.

“In the current study, we observed that the application of topical curcumin has given a protective effect against ultraviolet exposure-induced apoptosis in mice. However, this study has limitations because it did not examine markers of oxidative stress (reactive oxygen species, Hydrogen peroxide, and malondialdehyde) and inflammation (cyclooxygenase-2, interleukin, Prostaglandin E2, tumor necrosis factor-alpha, and nitric oxide) that may have an effect on the process of apoptosis.”

  1. In histopathological Fig 2A-B, it will be better if author can add effect after every treatment.

Response:

Thank you for your kind suggestion. We’ve revised the figure 2 legends.

“Figures 2. Apoptotic cells, indicated by brown staining in the nuclei (arrow), at 400x magnification under an Olympus Type CX-31 microscope, in UVB-Only Group with majority apoptosis cells (A); Acetone (B); 0.1µM Concentration (C); 1µM Concentration (D); 10µM Concentration; and 100µM Concentration with the least amount of cell apoptosis (F).”

  1. Arrow in the figure should be labelled properly.

Response:

Thank you for your kind suggestion. We’ve revised the arrow in the figure 2.

  1. Rest overall manuscript seems ok to me. Thanks good luck 

Response:

Thank you

Reviewer 2 Report (New Reviewer)

This original research article investigated the photoprotective effect of curcumin on mice subjected to UVB. The curcumin was applied daily or just once before UVB exposure. The findings proved the ability of this molecule to protect the skin and decrease apoptosis. Although there are some published works similar to this article, this field is still lacking research, especially on the mechanism of action of curcumin and how it protects the cells from UV radiation. It would've been an even more interesting article if more investigation was carried out.

Minor comments:

Keywords:  Please add more relevant keywords

Line 110-111: please add the type of microscope used.

Line 115: Please revise the font for "one way analysis of variance"

Table 1: avoid repeating curcumin twice in the title.

Figures 2A–B:  should be corrected with Figure 2. Also, please add in the title the type of microscope used. It would be better to provide the figures for the other concentrations to allow readers to have a complete microscopic observation.

Lines 212 - 215: this hypothesis is not clear. Please explain more since it is the explanation of your results.

Lines 215 - 17: what are these antioxidants? Please explain.

Line 229: in "cellular compounds, such as nucleic acids, proteins, and lipids, inflammation " what do you mean by inflammation? 

Author Response

Comments and Suggestions for Authors

Thank you for your valuable comment. Our replies to the reviewer's inquiries and revised points are as follows. Please confirm the relevant parts highlighted in yellow in the revised manuscript.

REVIEWER 2

This original research article investigated the photoprotective effect of curcumin on mice subjected to UVB. The curcumin was applied daily or just once before UVB exposure. The findings proved the ability of this molecule to protect the skin and decrease apoptosis. Although there are some published works similar to this article, this field is still lacking research, especially on the mechanism of action of curcumin and how it protects the cells from UV radiation. It would've been an even more interesting article if more investigation was carried out.

Minor comments:

  1. Keywords:  Please add more relevant keywords

Response:

Thank you for your kind suggestion. We’ve revised the keywords.

“Keywords: curcumin; inhibition; radiation; apoptosis.”

  1. Line 110-111: please add the type of microscope used.

Response:

We have revised the methods section on page 3, line 109-111.

“Then the tissues were rehydrated, immersed in xylol, covered with object glass, and examined under a microscope (Olympus Type CX-31, Tokyo, Japan).”

  1. Line 115: Please revise the font for "one way analysis of variance".

Response:

We have revised the methods section on page 3, line 116.

“…were one-way ANOVA for…”

  1. Table 1: avoid repeating curcumin twice in the title.

Response:

We have revised the table title on page 4, line 137-139.

“Table 1. Comparison in Apoptotic Cell Count Between One-Time and Seven-Time Curcumin Application Prior to Single 343 mJ UVB Exposure.”

  1. Figures 2A–B:  should be corrected with Figure 2. Also, please add in the title the type of microscope used. It would be better to provide the figures for the other concentrations to allow readers to have a complete microscopic observation.

Response:

Thank you for your kind suggestion. We’ve revised the figure 2 legends.

“Figures 2. Apoptotic cells, indicated by brown staining in the nuclei (arrow), at 400x magnification under an Olympus Type CX-31 microscope, in UVB-Only Group with majority apoptosis cells (A); Acetone (B); 0.1µM Concentration (C); 1µM Concentration (D); 10µM Concentration; and 100µM Concentration with the least amount of cell apoptosis (F).”

  1. Lines 212 - 215: this hypothesis is not clear. Please explain more since it is the explanation of your results.

Response:

We have revised the discussion section on page 6, line 215-220.

“However, we hypothesized that this phenomenon may be related to the biphasic effect of curcumin on the oxidation of post-prandial chylomicrons and its biphasic hormetic response on proteasome activity and heat-shock protein synthesis in human keratinocytes [20,21]. Furthermore, we postulated that the cumulative effect of antioxidants during the daily application of topical curcumin enhances the inhibitory effect on cell damage and apoptosis, as reported by Zhou, et al [22].”

  1. Lines 215 - 17: what are these antioxidants? Please explain.

Response:

We have revised the discussion section on page 6, line 215-223.

“However, we hypothesized that this phenomenon may be related to the biphasic effect of curcumin on the oxidation of post-prandial chylomicrons and its biphasic hormetic response on proteasome activity and heat-shock protein synthesis in human keratinocytes [20,21]. Furthermore, we postulated that the cumulative effect of antioxidants during the daily application of topical curcumin enhances the inhibitory effect on cell damage and apoptosis, as reported by Zhou, et al [22]. Another study regarding pre-treatment of oral curcumin during a seven-day period prevented cyclophosphamide-induced lung injury in rats through the suppression of oxidative stress, therefore reducing the number of cell apoptosis [23].”

  1. Line 229: in "cellular compounds, such as nucleic acids, proteins, and lipids, inflammation " what do you mean by inflammation?

Response:

We have revised the discussion section on page 6, line 234-236.

“…can damage various cellular compounds, such as nucleic acids, proteins, and lipids. It can lead to a gene mutation that acts as a precursor for photoaging and cutaneous malignancies.”

This manuscript is a resubmission of an earlier submission. The following is a list of the peer review reports and author responses from that submission.

Round 1

Reviewer 1 Report

The work is a scientific study about “The Role of Topical Curcumin in UVB-Induced Mice: Analysis of UVB-Induced Cell Apoptosis After Single and Daily Application” however needs some corrections or changes as I suggest:

1. In this manuscript, the author needs to maintain author instruction of the journal where the Abstract must be not maximum of 200 words.

2. Author should be considered punctuation problems throughout the manuscript.

3. Table 1 caption should be in detail.

4. From this available published literature, the novelty of this work is questionable.  The authors should properly explain the reason behind their results in this manuscript.

Whereas this area of research has already been covered by recent excellent articles published as follows:

Park, K., & Lee, J. H. (2007). Photosensitizer effect of curcumin on UVB-irradiated HaCaT cells through activation of caspase pathways. Oncology reports17(3), 537-540.

Dujic, J., Kippenberger, S., Hoffmann, S., Ramirez-Bosca, A., Miquel, J., Diaz-Alperi, J., ... & Bernd, A. (2007). Low concentrations of curcumin induce growth arrest and apoptosis in skin keratinocytes only in combination with UVA or visible light. Journal of Investigative Dermatology127(8), 1992-2000.

The results could be more significant and authentic if the author investigated the molecular-level study of apoptosis or DNA damage.

5. Author must follow the journal reference style.

Reviewer 2 Report

Please, refer to the attached file for comments. Thank you.

Reviewer 3 Report

Comments for Molecules-1954719

In the manuscript, “The role of topical curcumin in UVB-induced mice: analysis of UVB-induced cell apoptosis after single and daily application”, Djawad and co-workers proposed that curcumin, known as a natural antioxidant, may affect UVB-induced cutaneous malignancies. They used two UVB-induced mice models to detect apoptotic cells. They have concluded that daily topical curcumin can act as a potential photoprotective agent against UVB radiation. However, this research only started, and this paper is not ready for publication yet. 

Also, some suggestions are listed below for the authors’ consideration.

1)     Scientific question is not accurate.

2)     Paper writing is not the style of research. 

3)     Methods are simple and can not provide high-quality data.

4)     Result is rough and can not focus on scientific questions.

5)     Discussion is overreach, not focused on research.